# Stochastic Modelling of HIV-1 Replication in a CD4 T Cell with an IFN Response

**DOI:** 10.3390/v15020296

**Published:** 2023-01-20

**Authors:** Igor Sazonov, Dmitry Grebennikov, Rostislav Savinkov, Arina Soboleva, Kirill Pavlishin, Andreas Meyerhans, Gennady Bocharov

**Affiliations:** 1Faculty of Science and Engineering, Swansea University, Bay Campus, Fabian Way SA1 8EN, UK; 2Marchuk Institute of Numerical Mathematics of the RAS, 119333 Moscow, Russia; 3Moscow Center of Fundamental and Applied Mathematics at INM RAS, 119333 Moscow, Russia; 4World-Class Research Center “Digital Biodesign and Personalized Healthcare”, Sechenov First Moscow State Medical University, 119991 Moscow, Russia; 5Institute for Computer Science and Mathematical Modelling, Sechenov First Moscow State Medical University, 119991 Moscow, Russia; 6Department of Control and Applied Mathematics, Moscow Institute of Physics and Technology (National Research University), 141701 Dolgoprudny, Russia; 7Faculty of Computational Mathematics and Cybernetics, Lomonosov Moscow State University, 119991 Moscow, Russia; 8I CREA, Pg. Lluis Companys 23, 08010 Barcelona, Spain; 9Infection Biology Laboratory, Universitat Pompeu Fabra, 08003 Barcelona, Spain

**Keywords:** HIV life cycle, Type I interferon (IFN-I), viral dynamics, mathematical model, stochastic processes, Markov chain Monte Carlo method, sensitivity analysis

## Abstract

A mathematical model of the human immunodeficiency virus Type 1 (HIV-1) life cycle in CD4 T cells was constructed and calibrated. It describes the activation of the intracellular Type I interferon (IFN-I) response and the IFN-induced suppression of viral replication. The model includes viral replication inhibition by interferon-induced antiviral factors and their inactivation by the viral proteins Vpu and Vif. Both deterministic and stochastic model formulations are presented. The stochastic model was used to predict efficiency of IFN-I-induced suppression of viral replication in different initial conditions for autocrine and paracrine effects. The probability of virion excretion for various MOIs and various amounts of IFN-I was evaluated and the statistical properties of the heterogeneity of HIV-1 and IFN-I production characterised.

## 1. Introduction

Human immunodeficiency virus Type 1 (HIV-1) infections lead to the decline of immune functions followed by disease progression to AIDS [1]. Antiviral immune responses to HIV-1 are characterised by innate and adaptive immunity [2]. Studying of the host and viral factors that control viral replication and transmission is important to better understand the robustness and fragility of virus persistence.

The interferon Type I (IFN-I) response of innate immunity plays a central role in inhibiting viral replication [3]. The underlying processes include recognition of HIV-1 by innate immune sensors within a cell, induction of IFN-I production, and finally, the expression of antiviral genes (i.e., interferon-stimulated genes (ISGs)). However, the life cycle of HIV-1 also leads to the production of proteins (Vpu, Vif) that inhibit the effect of the antiviral proteins APOBEC3, Tetherin, and SAMHD1, resulting in resistance to IFN-I [4]. While the pathways regulating the production of IFN-I and the induction of ISGs are well studied [3], the quantitative kinetic description of these processes still needs to be developed.

The production of IFN-I starts at the initial infection and persists during disease progression [5]. However, this innate immune response is insufficient to suppress HIV-1 replication as the virus is able to evade the IFN-mediated antiviral activity. Sustained IFN-I expression bears deleterious effects on host immunity [6,7,8]. The mechanisms of the virus escaping intracellular defences are not completely understood [9,10,11] because they include multiple layers of control of IFN-I signalling [12] and diverse roles of interferon-stimulated gene (ISG) products [13].

We previously formulated a mathematical model describing the life cycle of HIV-1 in CD4 T cells [14]. In the present work, we extend the above model to consider the production of IFN-I by an infected cell, the ISG-mediating induction of effector molecules exhibiting anti-viral effects, and the inhibiting functions of viral proteins, as proposed in [15]. Our study of the stochastic effects during the HIV-1 life cycle suggests that the inherent randomness of reactions and the discrete nature of the biochemical stages of HIV-1 infection need to be considered in order to properly characterise the variability in viral production between infected cells. Furthermore, taking IFN-I into account and the proteins mediating viral resistance adds an additional layer of regulation for the heterogeneity of HIV-1 replication in a single cell. A dynamical model simulating the interaction of HIV-1 with IFNα at a cell population level was developed in [16]. A data-based single-cell model of the IFN-I response to Dengue virus infection was elaborated in [17].

The aim of the current work was to formulate and calibrate a stochastic model of HIV-1 replication that includes antiviral responses of an infected cell and to apply it to examine the dependence of virus production and IFN-I secretion on antiviral defence reactions. This requires the application of computational modelling tools, which corresponds to the mainstream trends in immunology and virology [12,18]. Specifically, we aimed to:
Develop an analytical tool for data assimilation and analysis of HIV-1 replication and the IFN-I response;Identify the control points underlying the viral ability to evade IFN-I-mediated defences as they represent potential targets for antiviral and immune therapies;Predict the effective reproduction number of single-cell infection resulting from the competition of multiple factors related to viral and IFN-I-dependent products.

In Section 2, we present a biochemical scheme of the processes that regulate HIV-1 production and IFN-I reactions. The equations of the deterministic model are described in detail and translated to a stochastic model using the Gillespie approach. The results of extensive numerical experiments with the model are presented in Section 3. The discussion and conclusions of the study can be found in Section 4 and Section 5, respectively.

## 2. Methods

### 2.1. Intracellular Type I Interferon Response and HIV-1 Replication

Upon HIV-1 infection of human cells, the virus is recognised by cytosolic sensors (RIG-1), which trigger a cascade of reactions, leading to the synthesis and activation of the IRF-3 and NF-kB proteins. The proteins translocate to the nucleus and cause the transcription of genes synthesising IFN-I [3,17,19].

Interferons are released into the extracellular space (IFNe) and bind to receptor molecules that are expressed on the cell surface of the produced cell or neighbouring cells. Binding of IFN-I to IFN-I receptors induces a signalling cascade with the participation of STAT-1 and STAT-2 proteins forming a transcription factor by assembling with IRF9. This translocates to the nucleus and initiates the transcription of ISGs. The following translation of the transcribed genes leads to the production of interferon-stimulated proteins such as APOBEC3, SAMHD1, and Tetherin. These proteins inhibit various stages of HIV-1 replication in the cell: APOBEC3 and SAMHD1 prevent reverse transcription, whereas Tetherin prevents the exit of the mature virion from the cell [19,20,21]. The action of ISGs is inhibited by the accessory proteins Vpu and Vif of HIV-1. These two viral proteins form complexes with the IFN-I-simulated antiviral proteins and target them for degradation [19,22]. Note that, as a result of the interaction of interferon-stimulated and HIV-1 accessory proteins, the amount of proviral DNA generated after reverse transcription decreases by about 50% in the presence of APOBEC3 [20] and by another 50% in the presence of SAMHD1 [23]. As observed in [21], Tetherin reduces the release of free virions by 40%.

### 2.2. Governing Deterministic Equations

The biochemical scheme if HIV-1 replication and the IFN-I response in a single cell is presented in Figure 1. The HIV-1 replication cycle consists of multiple stages, as described below. Based on this scheme, we developed a mathematical model of the HIV-1 life cycle in a CD4+ T cell. The major set of specific biochemical reactions and model parameters were taken from our previous work [14]. These were supplemented by a set of reaction stages for intracellular IFN-I synthesis and IFN-receptor-transduced responses of ISGs in an infected cell.

**Entry:** Free viral particles $V_{\mathrm{free}}$ are bound with the cell membrane according to the following equations: (1)$$\begin{array}{cc} {\dot{x}}_{1} & =-k_{\mathrm{bound}}x_{1}-d_{\mathrm{free}}x_{1} \end{array}$$(2)$$\begin{array}{cc} {\dot{x}}_{2} & =k_{\mathrm{bound}}x_{1}-k_{\mathrm{fuse}}x_{2}-d_{\mathrm{bound}}x_{2} \end{array}$$
where:$x_{1}=\left[V_{\mathrm{free}}\right]$ is the number of free virions in vicinity of the cell;$x_{2}=\left[V_{\mathrm{bound}}\right]$ is the number of virions bound to CD4 and the co-receptor.Here, $k_{\mathrm{bound}}=3.1$ h${}^{-1}$; $k_{\mathrm{fuse}}=0.7$ h${}^{-1}$; $d_{\mathrm{free}}=0.38$ h${}^{-1}$; $d_{\mathrm{bound}}=0.0008$ h${}^{-1}$.

**Reverse transcription and integration:** After the release of RNA following the entry of a virion into the cell, reverse transcription is initiated. Proteins SAMHD1 and APOBEC3 have an inhibitory effect on this initial step in viral replication. Then, the viral DNA penetrates the nucleus and is integrated into the chromosomal DNA of the host cell. Instead of integrating into the host chromosome as a provirus, the viral DNA can also undergo several circularisation reactions, thus losing the capability to support subsequent replication [24]. The respective model equations read: (3)$$\begin{array}{cc} {\dot{x}}_{3} & =k_{\mathrm{fuse}}x_{2}-k_{\mathrm{RT}}x_{3}-d_{{\mathrm{RNA}}_{\mathrm{cor}}}x_{3}-f_{\mathrm{APO}}x_{34}x_{3}-f_{\mathrm{SAM}}x_{35}x_{3} \end{array}$$(4)$$\begin{array}{cc} {\dot{x}}_{4} & =k_{\mathrm{RT}}x_{3}-k_{{\mathrm{DNA}}_{t}}x_{4}-d_{{\mathrm{DNA}}_{\mathrm{cor}}}x_{4} \end{array}$$(5)$$\begin{array}{cc} {\dot{x}}_{5} & =k_{{\mathrm{DNA}}_{t}}x_{4}-k_{\mathrm{int}}x_{5}-d_{{\mathrm{DNA}}_{\mathrm{nuc}}}x_{5} \end{array}$$(6)$$\begin{array}{cc} {\dot{x}}_{6} & =k_{\mathrm{int}}x_{5}-d_{{\mathrm{DNA}}_{\mathrm{int}}}x_{6} \end{array}$$
where:$x_{3}=\left[{\mathrm{RNA}}_{\mathrm{cor}}\right]$ is the number of genomic RNA molecules in the cytoplasm;$x_{4}=\left[{\mathrm{DNA}}_{\mathrm{cor}}\right]$ is the number of proviral DNA molecules synthesised by reverse transcription;$x_{5}=\left[{\mathrm{DNA}}_{\mathrm{nuc}}\right]$ is the number of DNA molecules in the nucleus;$x_{6}=\left[{\mathrm{DNA}}_{\mathrm{int}}\right]$ is the number of integrated DNA;$x_{34}=\left[\mathrm{APOBEC}3\right]$ is the number of molecules of the apolipoprotein B editing complex;$x_{35}=\left[\mathrm{SAMHD}1\right]$ is the number of molecules of the cellular enzyme, responsible for blocking the replication of HIV in dendritic cells, macrophages, monocytes, and resting CD4+ T lymphocytes.Here, $k_{\mathrm{RT}}=0.43$ h${}^{-1}$, $k_{{\mathrm{DNA}}_{t}}=0.12$ h${}^{-1}$, $k_{\mathrm{int}}=0.14$ h${}^{-1}$; $d_{{\mathrm{RNA}}_{\mathrm{cor}}}=0.21$ h${}^{-1}$, $d_{{\mathrm{DNA}}_{\mathrm{cor}}}=0.03$ h${}^{-1}$, $d_{{\mathrm{DNA}}_{\mathrm{nuc}}}=0.001$ h${}^{-1}$; $d_{{\mathrm{DNA}}_{\mathrm{int}}}=0.00002$ h${}^{-1}$; $f_{\mathrm{APO}}=0.35$ h${}^{-1}$, $f_{\mathrm{SAM}}=1.6$ h${}^{-1}$.The two last terms in Equation (3) describe the inhibitory effect of the SAMHD1 and APOBEC3 proteins.

**Transcription:** The process of transcription of full-length (mRNAg), single-spliced (mRNAss), and double-spliced (mRNAds) messenger RNA molecules takes place in the nucleus. All types of RNAs are then transported to the cytoplasm (mRNACg, mRNACss, mRNACds). The model equations for these reaction stages can be written as: (7)$$\begin{array}{cc} {\dot{x}}_{7} & =f_{\mathrm{TR}}\;x_{6}-k_{{\mathrm{ssRNA}}_{g}}g_{\mathrm{Rev}}x_{7}-k_{{\mathrm{eRNA}}_{g}}f_{\mathrm{Rev}}x_{7}-d_{{\mathrm{RNA}}_{g}}x_{7} \end{array}$$(8)$$\begin{array}{cc} {\dot{x}}_{8} & =k_{{\mathrm{ssRNA}}_{g}}g_{\mathrm{Rev}}\;x_{7}-k_{{\mathrm{dsRNA}}_{\mathrm{ss}}}g_{\mathrm{Rev}}x_{8}-k_{{\mathrm{eRNA}}_{\mathrm{ss}}}f_{\mathrm{Rev}}x_{8}-d_{{\mathrm{RNA}}_{\mathrm{ss}}}x_{8} \end{array}$$(9)$$\begin{array}{cc} {\dot{x}}_{9} & =k_{{\mathrm{dsRNA}}_{\mathrm{ss}}}g_{\mathrm{Rev}}\;x_{8}-k_{{\mathrm{eRNA}}_{\mathrm{ds}}}x_{9}-d_{{\mathrm{RNA}}_{\mathrm{ds}}}x_{9} \end{array}$$(10)$$\begin{array}{cc} {\dot{x}}_{10} & =k_{{\mathrm{eRNA}}_{g}}f_{\mathrm{Rev}}\;x_{7}-k_{\mathrm{tp},\mathrm{RNA}}x_{10}-d_{{\mathrm{RNA}}_{g}}x_{10} \end{array}$$(11)$$\begin{array}{cc} {\dot{x}}_{11} & =k_{{\mathrm{eRNA}}_{\mathrm{ss}}}f_{\mathrm{Rev}}\;x_{8}-d_{{\mathrm{RNA}}_{\mathrm{ss}}}x_{11} \end{array}$$(12)$$\begin{array}{cc} {\dot{x}}_{12} & =k_{{\mathrm{eRNA}}_{\mathrm{ds}}}x_{9}-d_{{\mathrm{RNA}}_{\mathrm{ds}}}x_{12} \end{array}$$
where:$x_{7}=\left[{\mathrm{mRNA}}_{g}\right]$ is the number of HIV mRNA molecules in the nucleus: g for genomic or full-length;$x_{8}=\left[{\mathrm{mRNA}}_{\mathrm{ss}}\right]$ is the number of HIV singly spliced (ss) mRNA molecules in the nucleus;$x_{9}=\left[{\mathrm{mRNA}}_{\mathrm{ds}}\right]$ is the number of HIV doubly spliced (ds) mRNA molecules in the nucleus;$x_{10}=\left[{\mathrm{mRNAc}}_{g}\right]$ is the number of HIV mRNA molecules in the cytoplasm: g for genomic or full-length;$x_{11}=\left[{\mathrm{mRNAc}}_{\mathrm{ss}}\right]$ is the number of HIV singly spliced (ss) mRNA molecules in the cytoplasm;$x_{12}=\left[{\mathrm{mRNAc}}_{\mathrm{ds}}\right]$ is the number of HIV doubly spliced (ds) mRNA molecules in the cytoplasm.Here, $k_{{\mathrm{ssRNA}}_{g}}=k_{{\mathrm{dsRNA}}_{\mathrm{ss}}}=2.4$ h${}^{-1}$, $k_{{\mathrm{eRNA}}_{g}}=k_{{\mathrm{eRNA}}_{\mathrm{ss}}}=2.3$ h${}^{-1}$, $k_{{\mathrm{eRNA}}_{\mathrm{ds}}}=4.6$ h${}^{-1}$, $k_{\mathrm{tp},\mathrm{RNA}}=2.8$ h${}^{-1}$, $d_{{\mathrm{RNA}}_{g}}=d_{{\mathrm{RNA}}_{\mathrm{ss}}}=d_{{\mathrm{RNAc}}_{\mathrm{ds}}}=0.12$ h${}^{-1}$.Functions $f_{\mathrm{TR}}\left(x_{13}\right)$, $f_{\mathrm{Rev}}\left(x_{14}\right)$, $g_{\mathrm{Rev}}\left(x_{14}\right)$ describe the Tat-Rev regulation of transcription:(12a)$$f_{\mathrm{TR}}={\mathrm{TR}}_{\mathrm{cell}}+\frac{x_{13}}{\theta_{\mathrm{Tat}}+x_{13}}{\mathrm{TR}}_{\mathrm{Tat}},\;f_{\mathrm{Rev}}=\frac{x_{14}}{\theta_{\mathrm{Rev}}+x_{14}},\;g_{\mathrm{Rev}}=1-\beta f_{\mathrm{Rev}}.$$Here, ${\mathrm{TR}}_{\mathrm{cell}}=15$ h${}^{-1}$, ${\mathrm{TR}}_{\mathrm{Tat}}=1500$ h${}^{-1}$, $\theta_{\mathrm{Tat}}=10^{3}$ molecules, $\theta_{\mathrm{Rev}}=77\times 10^{3}$ molecules, β=0.9.

**Translation of proteins:** Messenger RNA is decoded by the ribosomes to produce the Tat, Rev, Gag-Pol, Gag, gp160 proteins required for the production and assembly of new virions, as well as the auxiliary proteins Vpu and Vif, counteracting the response of the interferon system. The respective model equations are as follows: (13)$$\begin{array}{cc} {\dot{x}}_{13} & =k_{\mathrm{trans}}f_{\mathrm{ds},\mathrm{Tat}}x_{12}-d_{\mathrm{TaT}}x_{13} \end{array}$$(14)$$\begin{array}{cc} {\dot{x}}_{14} & =k_{\mathrm{trans}}f_{\mathrm{ds},\mathrm{Rev}}x_{12}-d_{\mathrm{Rev}}x_{14} \end{array}$$(15)$$\begin{array}{cc} {\dot{x}}_{15} & =k_{\mathrm{trans}}f_{g,\mathrm{Gag}\text{-}\mathrm{Pol}}x_{10}-k_{\mathrm{tp},\mathrm{Gag}\text{-}\mathrm{Pol}}x_{15}-d_{\mathrm{Gag}\text{-}\mathrm{Pol}}x_{15} \end{array}$$(16)$$\begin{array}{cc} {\dot{x}}_{16} & =k_{\mathrm{trans}}f_{g,\mathrm{Gag}}x_{10}-k_{\mathrm{tp},\mathrm{Gag}}x_{16}-d_{\mathrm{Gag}}x_{16} \end{array}$$(17)$$\begin{array}{cc} {\dot{x}}_{17} & =k_{\mathrm{trans}}f_{\mathrm{gp}160}x_{11}-k_{\mathrm{tp},\mathrm{gp}160}x_{17}-d_{\mathrm{gp}160}x_{17} \end{array}$$(18)$$\begin{array}{cc} {\dot{x}}_{18} & =k_{\mathrm{trans}}f_{\mathrm{ss},\mathrm{Vpu}}x_{11}-k_{\mathrm{tp}}x_{18}-d_{\mathrm{Vpu}}x_{18} \end{array}$$(19)$$\begin{array}{cc} {\dot{x}}_{19} & =k_{\mathrm{trans}}f_{\mathrm{ss},\mathrm{Vif}}x_{11}-d_{\mathrm{Vif}}x_{19} \end{array}$$
where$x_{13}=\left[\mathrm{Tat}\right]$, $x_{14}=\left[\mathrm{Rev}\right]$, $x_{15}=\left[\mathrm{Gag}\text{-}\mathrm{Pol}\right]$, $x_{16}=\left[\mathrm{Gag}\right]$, $x_{17}=\left[\mathrm{gp}160\right]$, $x_{18}=\left[\mathrm{Vpu}\right]$, and $x_{19}=\left[\mathrm{Vif}\right]$ are, respectively, the number of the Tat, Rev, Gag-Pol, Gag, gp160, Vpu, and Vif protein molecules.Here, $f_{g,\mathrm{Gag}\text{-}\mathrm{Pol}}=0.05$, $f_{g,\mathrm{Gag}}=0.95$, $f_{\mathrm{ss},\mathrm{gp}160}=0.64$, $f_{\mathrm{ds},\mathrm{Tat}}=0.025$, $f_{\mathrm{ds},\mathrm{Rev}}=0.2$, $f_{\mathrm{ss},\mathrm{Vpu}}=0.062$, $f_{\mathrm{ss},\mathrm{Vif}}=0.145$; $k_{\mathrm{trans}}=524$ h${}^{-1}$, $k_{\mathrm{tp},\mathrm{Gag}\text{-}\mathrm{Pol}}=k_{\mathrm{tp},\mathrm{Gag}}=k_{\mathrm{tp},\mathrm{gp}160}=k_{\mathrm{tp}}=2.8$ h${}^{-1}$;$d_{\mathrm{Gag}\text{-}\mathrm{Pol}}=d_{\mathrm{Gag}}=0.09$ h${}^{-1}$, $d_{\mathrm{gp}160}=0.02$ h${}^{-1}$, $d_{\mathrm{TaT}}=0.04$ h${}^{-1}$, $d_{\mathrm{Rev}}=0.07$ h${}^{-1}$, $d_{\mathrm{Vpu}}=0.39$ h${}^{-1}$, $d_{\mathrm{Vif}}=1.38$ h${}^{-1}$.

**Assembly of pre-virions at the membrane:** The Gag-Pol, Gag, and gp160 proteins and viral RNA molecules move to the membrane, where they form a pre-virion complex. We describe this stage using the following equations: (20)$$\begin{array}{cc} {\dot{x}}_{20} & =k_{\mathrm{tp},\mathrm{Gag}\text{-}\mathrm{Pol}}x_{15}-k_{\mathrm{comb}}N_{\mathrm{Gag}\text{-}\mathrm{Pol}}\;F_{c}\;x_{23}-d_{\mathrm{mem},\mathrm{Gag}\text{-}\mathrm{Pol}}x_{20} \end{array}$$(21)$$\begin{array}{cc} {\dot{x}}_{21} & =k_{\mathrm{tp},\mathrm{Gag}}x_{16}-k_{\mathrm{comb}}N_{\mathrm{Gag}}\;F_{c}\;x_{23}-d_{\mathrm{mem},\mathrm{Gag}}x_{21} \end{array}$$(22)$$\begin{array}{cc} {\dot{x}}_{22} & =k_{\mathrm{tp},\mathrm{gp}160}x_{17}-k_{\mathrm{comb}}N_{\mathrm{gp}160}\;F_{c}\;x_{23}-d_{\mathrm{mem},\mathrm{gp}160}x_{22} \end{array}$$(23)$$\begin{array}{cc} {\dot{x}}_{23} & =k_{\mathrm{tp},\mathrm{RNA}}x_{10}-k_{\mathrm{comb}}N_{\mathrm{RNA}}\;F_{c}\;x_{23}-d_{{\mathrm{RNA}}_{g}}x_{23} \end{array}$$(24)$$\begin{array}{cc} {\dot{x}}_{24} & =k_{\mathrm{tp},\mathrm{Vpu}}x_{18}-d_{\mathrm{Vpu}}x_{24} \end{array}$$(25)$$\begin{array}{cc} {\dot{x}}_{25} & =k_{\mathrm{comb}}\;F_{c}\;x_{23}-k_{\mathrm{bud}}x_{25}-d_{\mathrm{comb}}x_{25} \end{array}$$where: $x_{20}=\left[{\mathrm{Gag}\text{-}\mathrm{Pol}}_{\mathrm{mem}}\right]$, $x_{21}=\left[{\mathrm{Gag}}_{\mathrm{mem}}\right]$, $x_{22}=\left[{\mathrm{gp}160}_{\mathrm{mem}}\right]$, are, respectively, the number of the Gag-Pol, Gag, and gp160 viral protein molecules at the membrane;$x_{23}=\left[{\mathrm{RNA}}_{\mathrm{mem}}\right]$ is the number of viral RNA molecules at the membrane;$x_{24}=\left[{\mathrm{Vpu}}_{\mathrm{mem}}\right]$ is the number of Vpu protein molecules at the membrane;$x_{25}=\left[\mathrm{pre}\text{-}\mathrm{Virion}\right]$ is the number of pre-virion complexes at the membrane.
(25a)$$F_{c}=\frac{x_{20}}{x_{20}+K_{V_{\mathrm{rel}}}N_{\mathrm{Gag}\text{-}\mathrm{Pol}}}\cdot \frac{x_{21}}{x_{21}+K_{V_{\mathrm{rel}}}N_{\mathrm{Gag}}}\cdot \frac{x_{22}}{x_{22}+K_{V_{\mathrm{rel}}}N_{\mathrm{gp}160}},\;K_{V_{\mathrm{rel}}}=10^{3}.$$Here, $k_{\mathrm{comb}}=8.0$ h${}^{-1}$, $k_{\mathrm{bud}}=2.0$ h${}^{-1}$, $k_{\mathrm{tp},\mathrm{RNA}}=k_{\mathrm{tp},\mathrm{acc}}=k_{\mathrm{tp}}=2.8$ h${}^{-1}$;$N_{\mathrm{RNA}}=2$, $N_{\mathrm{Gag}\text{-}\mathrm{Pol}}=250$, $N_{\mathrm{Gag}}=5000$, $N_{\mathrm{gp}160}=24$, $d_{{\mathrm{RNA}}_{g}}=0.12$ h${}^{-1}$, $d_{\mathrm{mem},\mathrm{Gag}\text{-}\mathrm{Pol}}=d_{\mathrm{mem},\mathrm{Gag}}=0.004$ h${}^{-1}$, $d_{\mathrm{mem},\mathrm{gp}160}=0.014$ h${}^{-1}$, $d_{\mathrm{comb}}=0.52$ h${}^{-1}$.

**Budding and release of mature virions:** The virion buds and leaves the cell. The release process is hindered by the interferon-stimulated Tetherin protein. The accessory viral protein Vpu is transported to the membrane to capture the Tetherin molecules. These reactions are described by two equations: (26)$$\begin{array}{cc} {\dot{x}}_{26} & =k_{\mathrm{bud}}x_{25}-k_{\mathrm{mat}}x_{26}-f_{\mathrm{Teth}}x_{36}x_{26}-d_{\mathrm{bud}}x_{26} \end{array}$$(27)$$\begin{array}{cc} {\dot{x}}_{27} & =k_{\mathrm{mat}}x_{26}-d_{\mathrm{free}}x_{27} \end{array}$$
where:$x_{26}=\left[V_{\mathrm{bud}}\right]$ is the number of free viruses after budding from the cell;$x_{27}=\left[V_{\mathrm{mat}}\right]$ is the number of mature virions outside the cell;$x_{36}=\left[\mathrm{Tetherin}\right]$ is the number of Tetherin molecules.Here, $k_{\mathrm{mat}}=2.4$ h${}^{-1}$, $d_{\mathrm{bud}}=0.38$ h${}^{-1}$, $f_{\mathrm{Teth}}=0.008$ h${}^{-1}$.

**IFN synthesis activation in a cell:** The RIG-1 protein is activated by the viral RNA, starting a pathway activating the NF-kB and IRF3 proteins, which finally induce the synthesis of intracellular Type I interferon (IFNi). The respective equations of the model read: (28)$$\begin{array}{cc} {\dot{x}}_{28} & =k_{\mathrm{RIG}1}x_{3}-d_{\mathrm{RIG}}x_{28} \end{array}$$(29)$$\begin{array}{cc} {\dot{x}}_{29} & =k_{\mathrm{IRF}3}x_{28}-d_{\mathrm{IRF}3}x_{29} \end{array}$$(30)$$\begin{array}{cc} {\dot{x}}_{30} & =k_{\mathrm{NF}\text{-}\mathrm{kB}}x_{28}-d_{\mathrm{NF}\text{-}\mathrm{kB}}x_{30} \end{array}$$(31)$$\begin{array}{cc} {\dot{x}}_{31} & =k_{\mathrm{IFNi}}x_{29}+k_{\mathrm{IFNi}}x_{30}-k_{e}x_{31}-d_{\mathrm{IFNi}}x_{31} \end{array}$$
where:$x_{28}=\left[\mathrm{RIG}1\right]$ is the number of the RIG-1 protein molecules;$x_{29}=\left[\mathrm{IRF}3\right]$ and $x_{30}=\left[\mathrm{NF}\text{-}\mathrm{kb}\right]$ are the number molecules of the IRF3 and NF-kb proteins;$x_{31}=\left[{\mathrm{IFN}}_{i}\right]$ and $x_{32}=\left[{\mathrm{IFN}}_{e}\right]$ are the number of molecules of intercellular and extracellular IFN, respectively.Here, $k_{\mathrm{RIG}1}=0.48$ h${}^{-1}$, $k_{\mathrm{IRF}3}=1.02$ h${}^{-1}$, $k_{\mathrm{IFNi}}=1$ h${}^{-1}$, $k_{e}=0.13$ h${}^{-1}$, $k_{\mathrm{NF}\text{-}\mathrm{kB}}=0.91$ h${}^{-1}$, $d_{\mathrm{RIG}}=0.4$ h${}^{-1}$, $d_{\mathrm{IFNi}}=0.08$ h${}^{-1}$, $d_{\mathrm{IRF}3}=0.0015$ h${}^{-1}$, $d_{\mathrm{NF}\text{-}\mathrm{kB}}=0.00026$ h${}^{-1}$.

**Production of IFN stimulated proteins:** Interferon exits the cell (IFNe) and binds to the receptor on the membrane of other cells, activating the proteins STAT1 and 2, which start the production of the replication inhibitor proteins (APOBEC3, SAMHD1, Tetherin). The equations for APOBEC3, SAMHD1, and Tetherin take into account their loss due to the capture of molecules by auxiliary proteins of the virus and are specified as follows: (32)$$\begin{array}{cc} {\dot{x}}_{32} & =k_{e}x_{31}-d_{\mathrm{IFNe}}x_{32} \end{array}$$(33)$$\begin{array}{cc} {\dot{x}}_{33} & =k_{\mathrm{STAT}}x_{32}-d_{\mathrm{STAT}}x_{33} \end{array}$$(34)$$\begin{array}{cc} {\dot{x}}_{34} & =k_{\mathrm{ISG}}x_{33}-f_{\mathrm{Vif}}x_{19}x_{34}-d_{\mathrm{APO}}x_{34} \end{array}$$(35)$$\begin{array}{cc} {\dot{x}}_{35} & =k_{\mathrm{ISG}}x_{33}-d_{\mathrm{SAMHD}1}x_{35} \end{array}$$(36)$$\begin{array}{cc} {\dot{x}}_{36} & =k_{\mathrm{ISG}}x_{33}-f_{\mathrm{Vpu}}x_{23}x_{36}-d_{\mathrm{Teth}}x_{36} \end{array}$$
where:$x_{33}=\left[\mathrm{STAT}1,2\right]$ is the number of STAT-1–STAT-2 heterodimers;$x_{34}=\left[\mathrm{APOBEC}3\right]$ is the number of molecules of the apolipoprotein B editing complex;$x_{35}=\left[\mathrm{SAMHD}1\right]$ is the number of molecules of a cellular enzyme, responsible for blocking the replication of HIV in dendritic cells, macrophages, monocytes, and resting CD4+ T lymphocytes; $x_{36}=\left[\mathrm{Tetherin}\right]$ is the number of Tetherin molecules.Here, $k_{\mathrm{STAT}}=0.1$ h${}^{-1}$, $k_{\mathrm{ISG}}=2.94$ h${}^{-1}$, $d_{\mathrm{IFN}}=0.15$ h${}^{-1}$, $d_{\mathrm{STAT}}=0.03$ h${}^{-1}$, $d_{\mathrm{APO}}=0.087$ h${}^{-1}$, $d_{\mathrm{SAMHD}1}=0.16$ h${}^{-1}$, $d_{\mathrm{Teth}}=0.044$ h${}^{-1}$, $f_{\mathrm{Vif}}=f_{\mathrm{Vpu}}=7\times 10^{-6}$ h${}^{-1}$.

In the system of differential equations (ODE) above, 24 equations: (1)–(17), (20)–(23), and (25)–(27), coincide with those proposed in [14] and modified in [15] (disregarding two additional terms for the inhibition of proviral DNA synthesis in Equation (3)). The parameters in these equations were elaborated by Shcherbatova et al. [14] based on the analysis of published data (all necessary references are provided in [14]). In particular, the reaction rate constants of the HIV-1 transcription, translation, and assembly stages were estimated using two sources of data, i.e., the previously formulated mathematical models for the respective processes and the experimental data coming from quantitative studies of HIV-1 life cycle stages. One of the key experimental studies that provides a temporal description of HIV-1 replication [25] was used to refine the estimates of some parameters. It also describes the kinetics of the RNA, DNA, and viral proteins and, therefore, served as an overall validation of our model-reproduced dynamics.

Equations (18), (19), (24), and (28)–(36) are the ones describing the production of IFN-I and other proteins that take part in the inhibition of virus replication. We indicate here the publications in which the respective parameters were evaluated: $k_{\mathrm{RIG}1}$, $k_{\mathrm{IRF}3}$, $k_{\mathrm{IFNi}}$, $k_{e}$, $k_{\mathrm{STAT}}$, $d_{\mathrm{RIG}}$, $k_{\mathrm{ISG}}$, $d_{\mathrm{IFN}}$ [17]; $d_{\mathrm{IRF}3}$, $d_{\mathrm{STAT}}$ [26]; $d_{\mathrm{NF}\text{-}\mathrm{kB}}$ [27]; $d_{\mathrm{Vpu}}$ [28]; $d_{\mathrm{SAMHD}1}$ [29].

The RNA uptake rates by the APOBEC3 protein $f_{\mathrm{APO}}$ and by the Vif protein $f_{\mathrm{Vif}}$, as well as APOBEC3 degradation $d_{\mathrm{APO}}$ were estimated in [30]. The Vif protein degradation rate $d_{\mathrm{Vif}}$ was taken from [31]. A similar degradation rate for the SAMHD1 protein $f_{\mathrm{SAMHD}1}$, $d_{\mathrm{SAMHD}1}$ was quantified in [23]. The deactivation rate of the Tetherin protein by the Vpu protein $f_{\mathrm{Vpu}}$ was taken from [32]. Note that if the range of parameters is indicated in the work, we took the middle value. In some cases, we adjusted the value of the parameters (remaining in the indicated range) to reproduce the observed abundance of components.

Note that in [17,27], the data on the reaction components were specified in concentrations rather than the molecule numbers, i.e., [IRF3], [NF-kB], and [STAT1,2] are given in nM (nanomoles per litre), while intercellular and extracellular IFN are given in picograms per millilitre. Therefore, we provide scaling factors converting both units of concentrations into component abundances.

Conversion of concentration for:IRF3, [NF-kB], and [STAT1,2]: 1nM corresponds to $N_{A}\cdot 10^{-9}$ molecules per litre, where $N_{A}=6.022\cdot 10^{23}$ is the Avogadro number. The typical diameter of the CD4+ T cell is 6 μm, which gives its typical volume as $\Omega_{\mathrm{cell}}\approx 113\;\mu$m${}^{3}=113\cdot 10^{-15}$ L. Now, we can evaluate the factor $F_{\mathrm{nM}}=N_{A}\cdot 10^{-9}\times \Omega_{\mathrm{cell}}\approx 68$ molecules per nM.IFN${}_{i}$: 1 pg contains $N_{A}\cdot 10^{-12}/\mu_{IFN1}$ molecules, where $\mu_{IFN1}=19,500$ Da is the molar mass of IFN-I. Now, we can evaluate the factor $F_{\mathrm{IFNi}}=N_{A}\cdot 10^{-12}/\mu_{IFN1}\times \Omega_{\mathrm{cell}}\approx 0.0035$ molecules per pg/mL; here, $\Omega_{\mathrm{cell}}\approx 113\cdot 10^{-12}$ mL.IFN${}_{e}$: we assumed that the effective volume of the extracellular IFN molecules that can activate the STAT12 pathway is equal to the volume of the cell $\Omega_{\mathrm{cell}}$. This gives us the same terms for IFN export $k_{e}x_{31}$ in Equations (31) and (32) for computation as either concentrations or the number of molecules. Therefore, $F_{\mathrm{IFNe}}=F_{\mathrm{IFNi}}\approx 0.035$ molecules per pg/mL.

The deterministic model is described by a set of 36 ODEs. At time t=0, the initial conditions of the infection scenario were set as follows: initial number of free virions, $x_{1}\left(0\right)=\left[V_{\mathrm{free}}\right]$(0), and initial number of extracellular IFN, $x_{31}\left(0\right)=\left[{\mathrm{IFN}}_{e}\right]\left(0\right)$ molecules. All other components are zeros at t=0. As we are dealing with a single cell, the initial number of free virions $\left[{\mathrm{IFN}}_{e}\right]\left(0\right)$ coincides with the multiplicity of infection (MOI): the ratio of initial free virions to the number of cells. The corresponding initial-value problem can be integrated numerically, for example by the Runge–Kutta method. Examples of the integration results for MOI=6 and different values of $\left[{\mathrm{IFN}}_{e}\right]\left(0\right)$ are presented in Figure 2.

### 2.3. The Stochastic Model

#### 2.3.1. General Consideration

As shown in Figure 2, the abundance of some components, especially the first six of them, remained of the order of unity. Therefore, continuous trajectories for variables innately taking integer values (number of virions, molecules) are apparently not accurate. A natural way to account for the discreteness of the variables is the translation of the deterministic model in the form of an ODE system, to a stochastic description in the form of a Markov chain (MC). An efficient algorithm for such a translation was proposed by Gillespie [33,34,35]. It has been developed for chemical kinetic processes similar to those considered in viral replication stages.

To implement an MC numerically, a number of methods have been proposed with the most popular being Gillespie’s direct method [34]. In this method, we define the state vector $X={\left[x_{1},\ldots ,x_{N}\right]}^{T}$, where *N* is the number of components: N=36 in the case in hand.

Gillespie’s direct method can be briefly described as Algorithm 1. In this algorithm, $X_{0}$ is the initial state vector, which is the same as in the deterministic problem; only its component abundances must be integers.
**Algorithm 1:** Gillespie’s direct methodInitialise the state vector $X⇐X_{0}$ and time t⇐0;**while **$t<t_{\mathrm{final}}$**do**    Compute propensities $a_{m}\left(X\right),m=1,\ldots ,M$;    Compute the cumulative sum $A_{m}=\sum_{i=1}^{m}a_{i},m=1,\ldots ,M$;    Generate two uniformly distributed random numbers $r_{1},r_{2}\in \left[0,1\right]$;    Compute the time interval to fire the next reaction $\Delta t=-\ln\left(r_{1}\right)/A_{M}$;    Determine the index *m* of the next reaction: find the smallest *m*: $A_{m}>r_{2}\cdot A_{M}$;    Update time t⇐t+Δt;    Update the state vector X in accordance with Table 1;    Save t,X;**end while**

The solution of the MC converges in probability to the solution of the corresponding ODE system provided the proper scaling of all components and coefficients [36,37,38]. This limiting transition is called the fluid dynamics limit [36] or the mean field limit [39]. The theorem on the weak convergence of the MC process to the deterministic solution in application to viral infection dynamics was proven in [40,41].

#### 2.3.2. Specific Implementation

A Markov chain can be described as a list of elementary transitions (reactions) and their propensities. The propensity of the *m*th transition $a_{m}$ is defined as follows: $a_{m}\left(t\right)dt$ denotes the probability of the *m*th transition to occur in the interval [t,t+dt]. An important aspect of this approach is that it does not require additional quantification of the parameters, i.e., the deterministic model parameters of Equations (1)–(36) provide all the necessary information to be used for specifying the Markov chain. The Markov chain stochastic model corresponding to the underlying ODEs (1)–(36) is presented in Table 1. It contains M=81 transitions.

To obtain reliable statistical results with the MC description, a large number of realisations should be computed, which is time consuming. During the process, some components can reach large numbers. This essentially slows down the computation process, as a high population number causes extremely short time intervals between events in the Markov chain. At the same time, the evolution of highly populated components can be modelled rather accurately by a deterministic process described by an appropriate ODE. To overcome this issue, a hybrid approach was proposed in [38] in application to epidemic modelling and then developed in [15,40] in application to viral dynamics modelling.

In the developed version of the hybrid algorithm, stochastic and deterministic processes are running in parallel with the opportunity to switch automatically the dynamics of any component, $x_{n}$, from stochastic to deterministic and back as soon as this component exceeds a predefined threshold $\bar{X}$ or, respectively, becomes below it. Therefore, at any time interval between the transitions, all components are divided into two time-varying sets: $S_{X}=\left\{n:x_{n}\leq \bar{X}\right\}$ and $\bar{S_{X}}=\left\{n:x_{n}>\bar{X}\right\}$. Components, $x_{n}\in S_{X}$, having currently a relatively small number of particles are modelled stochastically by the Markov chain. Other components, $x_{n}\in \bar{S_{X}}$, with a large population size are modelled by the corresponding deterministic ODEs. With the change of population, a component, $x_{n}$, can be moved automatically from one set to another.

To accelerate the computations, the binary search method was employed for determining the next reaction index, *m*. Furthermore, special auxiliary arrays were prepared, which allowed updating only the components involved in the reaction performed and updating only the propensities affected by the updated components without spending time on other reactions. As the algorithm was implemented in C++, the arrays of pointers to functions were used to call the functions of the propensities and the ODE right-hand sides that have to be calculated for a given reaction directly. The computations were run on an Intel Xeon E3-1220 v5 CPU 3GHz×4.

## 3. Results

### 3.1. Sensitivity of HIV-1 and IFN-I Secretion to IFN-Mediated Control

The focus of our study was on the analysis of the IFN-I response effect on HIV-1 replication. The model was built upon our previous work [14], in which the sensitivity of the HIV-1 life cycle to all considered biochemical reaction steps was thoroughly examined. In order to keep the focus of the present work on IFN-I function, we restricted the sensitivity examination using heat maps, uncertainty bands, and PDFs to characterise IFN-I secretion and the IFN-I-induced function of ISG-produced proteins inhibiting virus replication.

The kinetic parameters of the reaction cascade from the activated IFN receptor to ISGs determine the degree of Type I IFN-mediated protection. The values of the parameters estimated are characterised by uncertainty. Therefore, the analysis of the variation in the numbers of released virions and synthesised IFN-I (both internal and external) was performed. The results of the sensitivity analysis of the deterministic model solution (number of released virions $\left[V_{\mathrm{mat}}\right]$ and the number of intracellular $\left[{\mathrm{IFN}}_{i}\right]$ and intercellular $\left[{\mathrm{IFN}}_{e}\right]$ molecules of IFN at t=36 h) with respect to variations in parameters $f_{\mathrm{Teth}}$, $k_{\mathrm{STAT}}$, $k_{\mathrm{IFNi}}$, and $k_{\mathrm{ISG}}$ are presented in Figure 3 in the form of heat maps. The heat maps demonstrate the effectiveness of various processes underlying the IFN-mediated suppression of the virus replication in a cell in the absence of the IFN signal from the other cells (i.e., the autocrine mode of control). The dependence of the secreted HIV-1 on the parameters considered demonstrates a synergistic effect, i.e., the effect of a joint variation is stronger than that of the individual parameters. The results suggest that a simultaneous increase in the rates of IFN induction $k_{\mathrm{IFNi}}$ and STAT1 and 2 pathway activation $k_{\mathrm{STAT}}$ is the most effective way of boosting the IFN antiviral protection of the host cell. As one can expect, the selected parameters related to the ISGs activation path do not impact the secretion of $\left[{\mathrm{IFN}}_{i}\right]$ and intercellular $\left[{\mathrm{IFN}}_{e}\right]$. However, the parameter defining the production of IFN-I, i.e., $k_{\mathrm{IFNi}}$, demonstrates a strong impact on the number of internal and external molecules.

### 3.2. Stochastic Dynamics

Representative examples of stochastic trajectories are presented in Figure 4 for all 36 components. These realisations were calculated for the initial conditions: MOI=6 and $\left[{\mathrm{IFN}}_{e}\right]\left(0\right)=0$. Here, the deterministic trajectories are shown by black solid curves. One can observe that the stochastic trajectories deviate essentially from the deterministic curves. This means that random fluctuations in the reaction rates and low numbers of the reaction species result in essentially the stochastic dynamics of the viral replication.

The meaning of the colour code for the stochastic trajectories can be worked out from Figure 5, where the kinetics of virion release from the cell is shown. The number of all new virions produced and released by an infected cell, [V${}_{\mathrm{new}}$], is computed by integrating the first term on the right-hand side of Equation (27), i.e., neglecting the degradation of released virions outside the cell:(37)$$\left[V_{\mathrm{new}}\right]\left(t\right)=\int_{0}^{t}k_{\mathrm{mat}}\left[V_{\mathrm{bud}}\right]\left(t^{\prime }\right)\;dt^{\prime }.$$The trajectory colours were selected by the following rule: the larger the released progeny, the closer is the colour to the red end of the spectrum, and vice versa, the lower the output, the closer is the colour towards the blue end of the spectrum.

This enables tracing back the trajectories through all components with the higher (red and orange) and lower (green and blue) number of all released virions during the process to identify the degree of coordination of individual reaction steps. We can see that, already beginning with $x_{4}=\left[{\mathrm{DNA}}_{\mathrm{cor}}\right]$, the red lines are on top. Hence, the fluctuations in the number of released virions are determined at the earlier stages of the infection process. However, the ensembles of realisations for components [IRF3], [NF-kB], [IFN${}_{i}$], [IFN${}_{e}$], and [STAT1,2] have a mixed colour structure. Additionally, the trajectories for [IRF3] and [NF-kB] have approximately the same shape, reminiscent of logistic curves, i.e., a fast growth at the early stage, followed by almost constant persistence.

Analysing the blue lines in the plots for [APOBEC3] and [Tetherin] in Figure 4, we observe that the extinct and low-output realisations are characterised by high numbers of these proteins at the late stages of the HIV-1 replication process.

Interestingly, some trajectories are clustered around certain values. This is clearly noticeable for the components [mRNAg], [mRNAc${}_{g}$], [Gag-Pol], and [Gag], but the tendency for similar clusterisation can be observed also for [mRNAc${}_{\mathrm{ss}}$], [g160], [Vpu], [Vif], [Vpu${}_{\mathrm{mem}}$], and [RNA${}_{\mathrm{mem}}$].

Some realisations with blue colour in Figure 5 having zero values of [V${}_{\mathrm{new}}]$ are degenerated or extinct realisations in which the infection process is not developed and new virions are not released.

### 3.3. Structural Analysis of PDF for Stochastic Realisations

In Figure 6, the time-varying histograms reflecting the probability density function (PDF) of the stochastic realisation are presented for 36 components or the same initial conditions: MOI=6 and $\left[{\mathrm{IFN}}_{e}\right]\left(0\right)=5$. The deterministic trajectories, as well as the curves for the mean and median values are also plotted with the colour code being explained in the legend. A darker colour corresponds to a higher value of the histogram. Most histograms are far from having a simple shape. Several maxima in the histogram are noticeable for [mRNA${}_{g}$], [mRHA${}_{\mathrm{ss}}$], [mRNAc${}_{g}$], [mRNAc${}_{\mathrm{ss}}$], [Gag-Pol], [Gag], [gp160], and [Vif]. Structural patterns in the trajectory ensembles are visible in the histograms for all the model components (although to a various degree) in Figure 6.

The time-dependent evolution of the histograms for [V${}_{\mathrm{mat}}$] is depicted in Figure 7 for MOI=6 and for different values of $\left[{\mathrm{IFN}}_{e}\right]\left(0\right)=0,5,10$ molecules. Here, a darker colour corresponds to a higher value of the PDF related to the histogram. Several maxima are also noticeable in the histograms, so that the lower is the value of the initial extracellular IFN-I, the larger is the number of noticeable maxima.

### 3.4. Identifying the PDFs of Stochastic Dynamics

The large number of stochastic realisations of the model provides the possibility to access the PDFs of the stochastic trajectories. In Figure 8 are shown the histograms for mature virions (left) and intercellular IFN (right) at the final time t=36 h for MOI=6 and different values of $\left[{\mathrm{IFN}}_{e}\right]\left(0\right)$ (explained in the legend). The histograms are normalised by the number of realisations in order to approximate the probability distribution function (PDF). The mean values at t=36 h are indicated by vertical dashed lines with the corresponding colour. Both distributions shown in Figure 8 are far from the normal one, which indicates the complexity of the emergent patterns of the stochastic viral infection dynamics.

In Figure 8 (left), the histogram has a certain number of bell-shaped peaks. In the case $\left[{\mathrm{IFN}}_{e}\right]\left(0\right)=0$, the number of noticeable peaks is larger; however, the histogram is smoother, and the peaks slightly exceed the mean value of released virions. The highest peak is not the first one. When external IFN-I is considered (the blue and green curves), the peaks are more pronounced, but the histogram decays faster on the right. The lower is the amount of initial extracellular IFN-I, the closer to the origin is the location of the first highest peak.

Figure 8 (right) shows the PDFs for the amount of intercellular IFN-I produced during the life cycle of HIV-1. The PDFs are close to the exponential distribution for $\left[{\mathrm{IFN}}_{e}\right]\left(0\right)>0$, whereas the PDF for $\left[{\mathrm{IFN}}_{e}\right]\left(0\right)=0$ has a bell-shaped peak. All PDFs seem to demonstrate exponential tails.

### 3.5. Heterogeneity of the Size of HIV-1 Progeny

To examine the natural variability of the infection process in a cell, it is appropriate to deal with the total number of released virions during the process:(38)$$\left[V_{\mathrm{tot}}\right]=\int_{0}^{T}k_{\mathrm{mat}}\left[V_{\mathrm{bud}}\right]\left(t^{\prime }\right)\;dt^{\prime }.$$

In our study, we restricted our consideration of the duration of the viral replication cycle by T=36 h, as approximately after 36 h, the CD4+ T cell dies [42]. Therefore, we set here
(39)$$\left[V_{\mathrm{tot}}\right]\approx \left[V_{\mathrm{new}}\right]\left(36\;h\right).$$

Normalised histograms (PDF) for [V${}_{\mathrm{tot}}$] are shown in Figure 9 for different numbers of initial free virions infecting the cell and different numbers of extracellular IFN-I molecules. They resemble the histogram for [V${}_{\mathrm{mat}}$] shown in Figure 8 (left), demonstrating multiple peaks with the peak amplitude decaying faster with the number of produced virions. The inspection of the figures shows that the right PDF tails are Gaussian. The greater is the initial number of virions (MOI), the lower is the amplitude of the first peak. Note that, for MOI=8 and in the absence of the extracellular IFN-I (Figure 9 (right)), the individual peaks are almost indistinguishable. However, the second peak looks to be the highest one. For MOI=6 and zero amount of extracellular IFN-I, the first two peaks have a similar value.

### 3.6. Uncertainty Bands in the Dynamics of HIV-1 and IFN-I Production

For many practical applications, it is instructive to evaluate the confidence intervals. The time evolution of the confidence intervals for $\left[V_{\mathrm{new}}\right]\left(t\right)$ is shown in Figure 10 for different initial conditions, i.e., the number of free virions attached to the cell at t=0 (denoted in the figure as MOI) and the initial number of external IFN-I molecules activating the ISG pathway (denoted in the figures as IFN).

The 25–75% confidence intervals (which include 50% of all realisations) are marked by yellow patches. The 15–85% confidence intervals are shown by the light-blue patches. They overlap with the 25–75% confidence intervals. The widest 5–95% confidence intervals (which include 90% of all realisations) are shown by the light-pink patches. The coloured patches in Figure 10 provide quantitative details of the evolution of the histograms for the total number of released virions during the development of the infection process. The deterministic trajectories, as well as the evolution curves for the mean value and median of $\left[V_{\mathrm{free}}\right]\left(t\right)$ are also depicted in Figure 10. Observe that, for $\left[V_{\mathrm{free}}\right]\left(0\right)=4$ and in presence of extracellular IFN-I in the amount of 10 molecules, the median is identically zero (left bottom). This means that, in more than 50% of cases, the stochastic replication process is extinct, so that new virions are not released.

The number of total virions, $\left[V_{\mathrm{tot}}\right]$, released during the infection process for different values of MOI and [IFN${}_{e}$](0) are presented in Table 2 for MOI=4,6,8 and $\left[{\mathrm{IFN}}_{e}\right]\left(0\right)=0,5,10$ molecules. Here, one can see the values predicted by the deterministic model (the corresponding columns are denoted as “det”) and the mean values obtained in the stochastic approach (the corresponding columns are denoted as “mean”). As specified in Table 2, the mean value of $\left[V_{\mathrm{tot}}\right]$ always exceeds the value obtained by the deterministic model. The percentage of excess is indicated in the column denoted by Δ.

### 3.7. HIV-1 Life Cycle Efficiency

The efficiency of an HIV-1 infection can be characterised by the ratio of the total viral progeny to the number of free virions infecting the cell (MOI). We define this value as the life cycle efficiency, similar to [43]:(40)$$\mathrm{Life}\;\mathrm{Cycle}\;\mathrm{Efficiency}=\frac{\left[V_{\mathrm{tot}}\right]}{\mathrm{MOI}}.$$The estimated values of the life cycle efficiency are indicated in the corresponding rows of Table 2.

To characterise the suppression of viral production by IFN-I, the values in the rows denoted as “Inhibitory Factor” are presented, quantifying the inhibitory effect of extracellular IFN. They are estimated as the ratio of the total number of released virions in the presence of IFN-I to the case when the extracellular IFN is absent as follows:(41)$$\mathrm{IFN}\;\mathrm{Inhibitory}\;\mathrm{Factor}=\frac{\left[V_{\mathrm{tot}}\right]\mathrm{at}\left[{\mathrm{IFN}}_{e}\right]\left(0\right)=0}{\left[V_{\mathrm{tot}}\right]\mathrm{at}\left[{\mathrm{IFN}}_{e}\right]\left(0\right)>0}.$$

The results presented in Table 2 show that, the smaller the initial number of free virions and the higher the extracellular IFN concentration, the more the stochastic mean value exceeds the deterministic one, i.e., by a factor of 2.5 for MOI=4 and $\left[{\mathrm{IFN}}_{e}\right]\left(0\right)=10$. At the same time, for $\left[{\mathrm{IFN}}_{e}\right]\left(0\right)=0$, the difference between the deterministic and mean stochastic outputs is smaller and close to 15%. This can also be seen in the plots shown in Figure 11 (left), where the kinetics of the deterministic model estimates [V${}_{\mathrm{new}}$] and mean values predicted by the stochastic model are depicted. This feature can be explained by the fact that the smaller the number of particles, the higher are the relative fluctuations, caused by stochasticity of the replication–inhibition process. Interestingly, the IFN-I inhibitory factor turns out to be lower in the stochastic model, particularly at a smaller number of free virions (MOI).

As mentioned above, the stochastic realisations with zero number of newly produced virions, i.e., $[V_{\mathrm{tot}}]=0$, are called degenerate or extinct. The developed stochastic model enables computing the probability of extinct cases, $P_{e}$, in relation to the initial number of free virions (MOI) and the amount of extracellular IFN-I molecules. The corresponding results for the probabilities of productive infection of a target cell, which is $1-P_{e}$, are shown in Figure 11 (right). New virions are produced in more than 50% of the realisations for MOI=2 only if $\left[{\mathrm{IFN}}_{e}\right]\left(0\right)=0$. In the presence of extracellular IFN, larger MOIs are needed to ensure a productive infection, i.e., MOI=4 for $\left[{\mathrm{IFN}}_{e}\right]\left(0\right)=5$ and MOI=7 for $\left[{\mathrm{IFN}}_{e}\right]\left(0\right)=10$. This dependence indicates the sensitivity of inhibitory effects of extracellular IFN-I.

## 4. Discussion

The treatment of HIV-1 infection in humans is based on highly active antiretroviral therapy [44]. Combination approaches that enhance the immune control are still under investigation [45]. IFN-I refers to a group of key host proteins that activate intracellular defence processes and inhibit HIV-1 replication. However, the clinical application of IFN-I has not yet demonstrated the expected medical benefit [3,44]. The lack of knowledge about the interactions between the activation and inhibition of antiviral immunity can be one of factors explaining this fact [46]. The design of multi-modal therapies of HIV-1 infection requires a predictive understanding of their regulation. This can be achieved by mathematical modelling both at the single-cell level and at the level of innate and adaptive immune responses [16].

In this study, we developed a mathematical model integrating the HIV-1 life cycle in a CD4 T cell with the activation of IFN-I responses and the resulting inhibition of the viral replication, which is in turn is suppressed by viral proteins (Vpu, Vif). Both the deterministic and the stochastic formulations were presented.

The stochastic model was employed to predict the efficiency of IFN-I-induced suppression of viral replication in both autocrine and paracrine mechanisms. The probabilities of productive infection for various MOIs were estimated. As a measure of the efficiency of viral replication in a single cell in the framework of the stochastic model, the mean value of $\left[V_{\mathrm{tot}}\right]$ was used. Indeed, the total viral progeny secreted by an ensemble of infected cells in lymphoid tissue should be calculated by summing new virions produced by every cell. Then, it is the mean value of the total viral progeny per available target cell that is directly related to the MOI, considered in the model. Hence, the estimates defined by Equations (40) and (41) based on $\left[V_{\mathrm{tot}}\right]$ are proper characteristics of the HIV-1 replication and the IFN inhibitory effect, respectively, provided that the mean value of $\left[V_{\mathrm{tot}}\right]$ is used in (40) and (41).

The analysis of the stochastic model suggests that the level of HIV-1 Gag protein at the membrane of the infected cell, required for the assembly of the pre-virion complex, is a limiting factor. Therefore, it can be considered as a potential target for drug therapy. It has also been found that an infected CD4 T cell produces a small amount of internal interferon, which is not enough to counteract the virus replication cycle. Indeed, the main producers of Type I IFN are the plasmacytoid dendritic cells (pDCs) [44]. Hence, the protection of CD4 T cells depends on the availability of extracellular IFN-I. The effects of various numbers of IFN-I molecules on the net viral progeny were quantified.

The model predicted that the life cycle efficiency does not change with a two-fold increase of the MOI, i.e., from 4 to 8. On the contrary, the availability of the extracellular IFN-I on a per-cell basis from 0 to 10 molecules reduces viral production three times. Thus, one can hypothesise that the inability of the IFN-I response to block early viral spreading in human infections is due to insufficient amounts of extracellular IFN-I compared to the number of available target cells. This corroborates a previously computationally predicted phenomenon of a diffusion-mediated compartmentalisation of IFN-I in lymph nodes [47] recently confirmed for antiretrovirals by quantitative imaging analysis [48].

In terms of the molecular virology of HIV-1 replication, the developed model predicts that continuous activation of STAT signalling and ISG products does not effect the net production of IFN-I by infected cells. Nevertheless, this activation along with the Tetherin production has a strong impact on the replication of viral particles. The activation rate constants of STAT-mediated signalling, interferon production, and the action of Tetherin predict a strong synergistic effect on the reduction of viral production. These model-derived predictions can be used in the data assimilation and analysis of recent assays for the HIV-1 life cycle [49,50].

Infected cell apoptosis was not considered in the model, as it would require the consideration of a whole network of regulatory processes and, thus, would complicate essentially the developed model. However, the model can be upgraded in the future to include the regulation of cell death.

Overall, our study provides the next step towards a mechanistic description and prediction of the interplay between HIV-1 replication, autocrine and paracrine responses of IFN-I, and resistance to IFN mediated by viral proteins. Future modelling works will include the effects of IFN-I on the physiology of infected CD4 T cells, thus providing an in silico tool to study a delicate balance “...that tips IFN response from friend to foe” [2].

## 5. Conclusions

A stochastic model of the HIV-1 life cycle in CD4 T cells and the reaction of the interferon system was developed. The reaction rates of the biochemical process network were calibrated. A range of heterogeneity characteristics of HIV-1 replication and the impact of IFN-I-mediated control were quantified. The model can be used for predicting the effect of IFN-I on viral replication for various MOI. The stages of the HIV-1 life cycle and the response of the interferon system in an infected cell were examined to identify the sensitive steps. Overall, we developed an analytical tool for the assimilation and analysis of data on HIV-1 replication and IFN-I responses. Using the computational model, the key control points underlying the ability of the virus to evade IFN-I-mediated defences and potential targets for antiviral drugs in combination with immune therapies were specified. The model predicted the effective reproduction number, i.e., the productive infection versus non-productive infection, at a single-cell level resulting from the competition of multiple factors related to viral and IFN-I-dependent proteins.

## Figures and Tables

**Figure 1 viruses-15-00296-f001:**
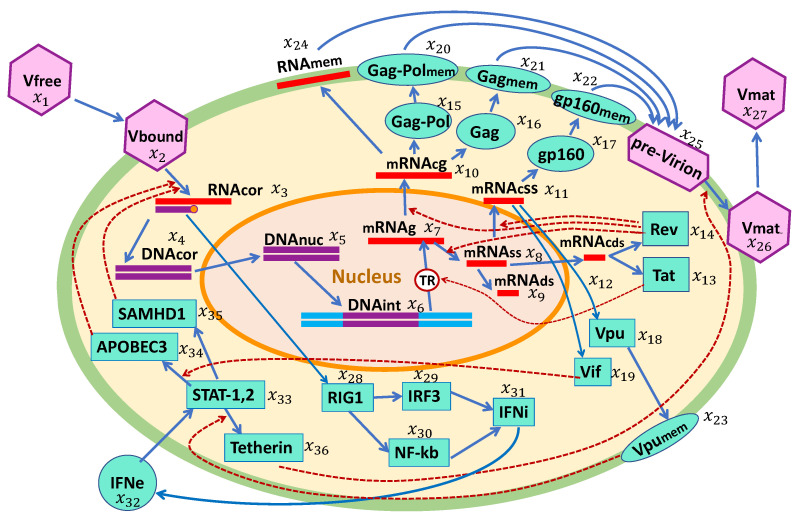
Biochemical scheme of the HIV-1 replication cycle in the presence of an IFN-I response. The consecutive chain of elementary processes comprises: viral entry, reverse transcription, integration into the chromosome, transcription and splicing of viral RNAs, translation of proteins including the proteins inhibiting the action of ISGs, assembly of pre-virions, budding and release of mature virions, sensing of viral RNAs, IFN synthesis, and translation of antiviral proteins by IFN-stimulated genes.

**Figure 2 viruses-15-00296-f002:**
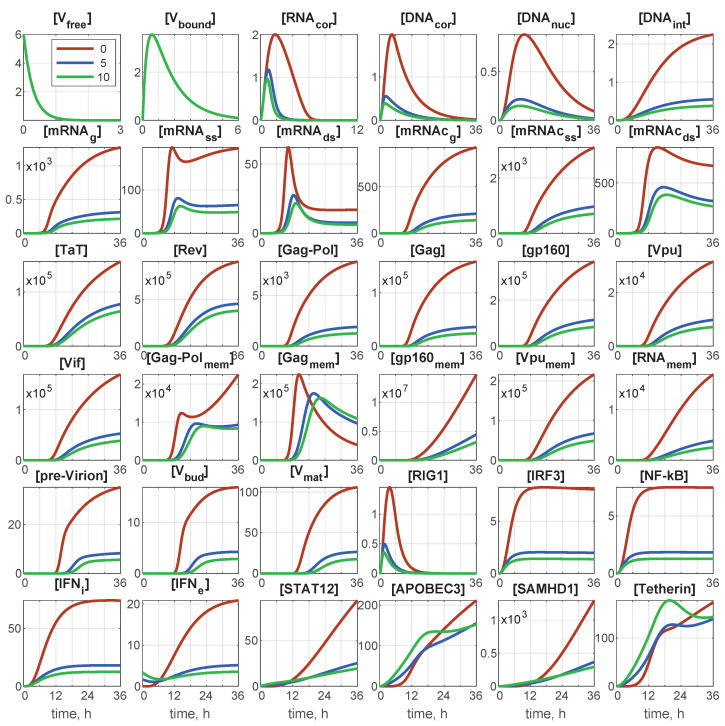
Deterministic trajectories for all components for MOI=6 and for different values of initial interferon $\left[{\mathrm{IFN}}_{e}\right]\left(0\right)$. Red line: 0 molecules; blue line: 5 molecules; green line: 10 molecules. Note that all components are indicated through the number of particles/virions/molecules.

**Figure 3 viruses-15-00296-f003:**
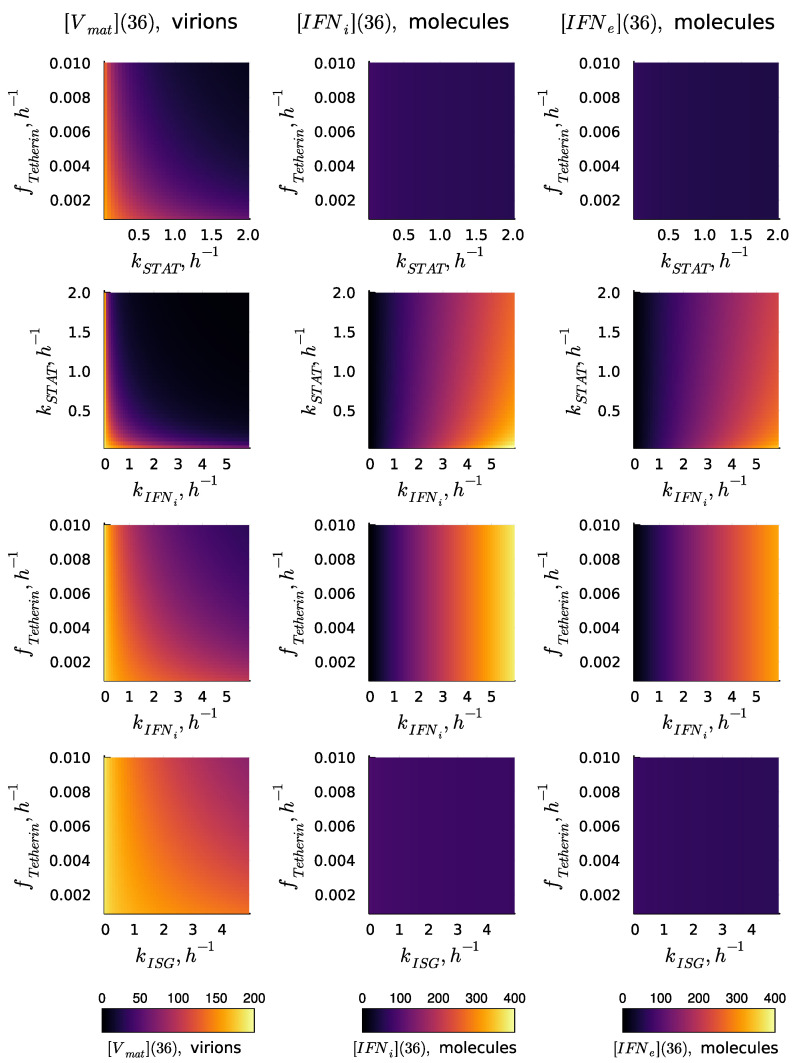
Effectiveness of various processes underlying IFN-mediated suppression of virus replication in a cell in the absence of IFN from other cells, i.e., the autocrine mode of control. Results of the sensitivity analysis of the deterministic model solution ($\left[V_{\mathrm{mat}}\right]\left(36\right)$, [IFNi](36), and $\left[{\mathrm{IFN}}_{e}\right]\left(36\right)$) with respect to variations in the parameters $f_{\mathrm{Tetherin}}$, $k_{\mathrm{STAT}}$, $k_{\mathrm{IFNi}}$, and $k_{\mathrm{ISG}}$. The results correspond to the MOI of $\left[V_{\mathrm{free}}\right]\left(0\right)=6$ virions without extracellular IFN signalling: $\left[{\mathrm{IFN}}_{e}\right]\left(0\right)=0$.

**Figure 4 viruses-15-00296-f004:**
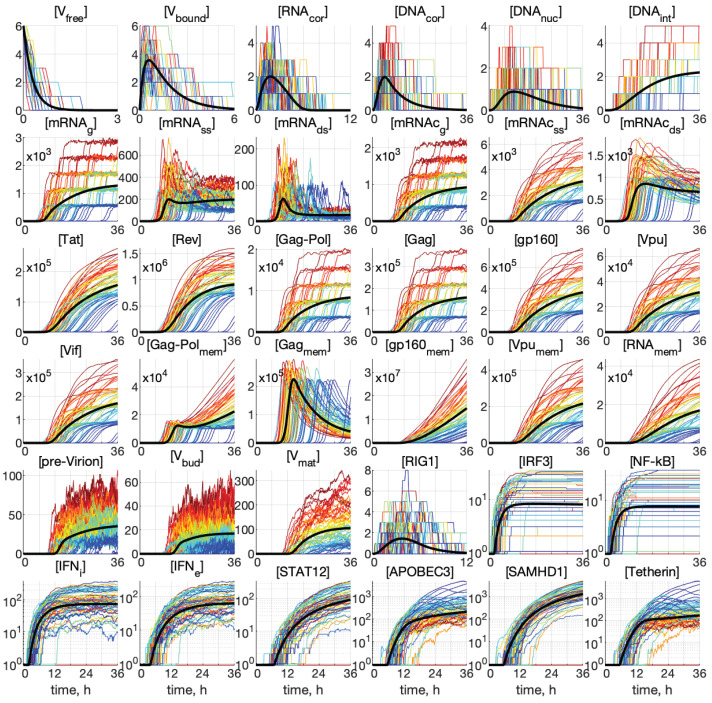
Examples of stochastic realisations for all 36 components for MOI=6 and $\left[{\mathrm{IFN}}_{e}\right]\left(0\right)=0$. The black curves indicate deterministic trajectories. The larger the released progeny, the closer is the colour to the red end of the spectrum, and vice versa, the lower the output, the closer is the colour towards the blue end of the spectrum. The stochastic trajectories deviate essentially from the deterministic curves, indicating that random fluctuations in the reaction rates and low numbers of the reaction species result in essentially heterogeneity in the viral replication components.

**Figure 5 viruses-15-00296-f005:**
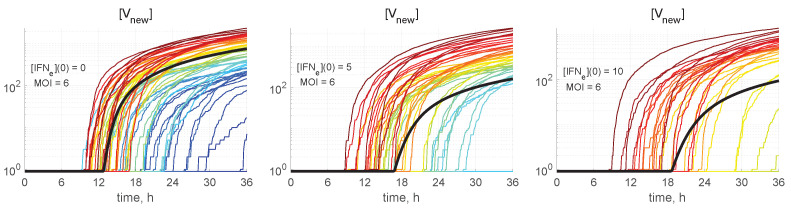
The number of all new virions produced and released by an infected cell in relation to the abundance of external IFN. Stochastic trajectories for $\left[V_{\mathrm{new}}\right]\left(t\right)$ and different values of initial intercellular IFN, indicated in every plot. The bold black line represents the corresponding deterministic trajectory. The larger the released progeny, the closer is the colour to the red end of the spectrum, and vice versa, the lower the output, the closer is the colour towards the blue end of the spectrum.

**Figure 6 viruses-15-00296-f006:**
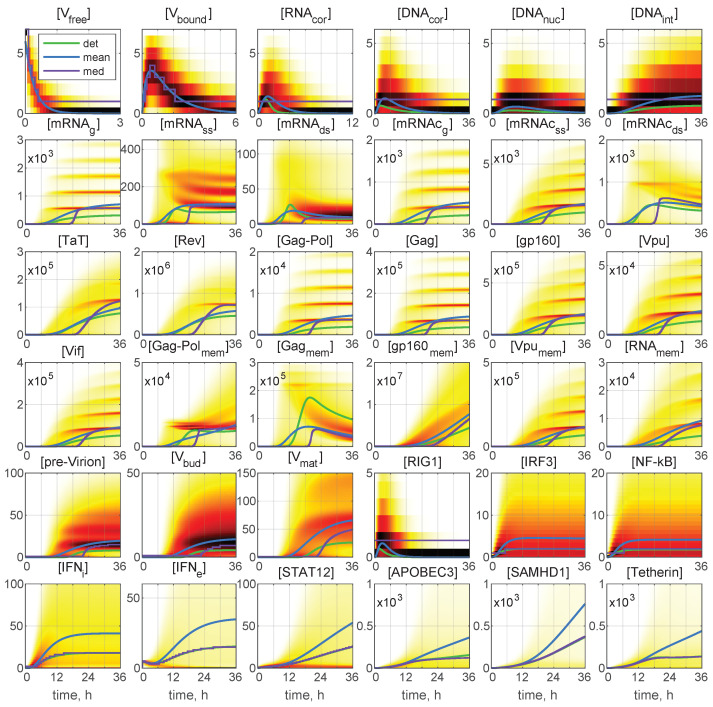
Probability density function (PDF) of stochastic realisations. Sliding histograms for 36 components for MOI=6 and $\left[{\mathrm{IFN}}_{e}\right]\left(0\right)=5$. Furthermore, the deterministic solutions (det), mean values (mean), and medians (med) are plotted (colours for the lines are explained in the legend). A darker colour corresponds to a higher value of the histograms. Multi-hump patterns in the trajectory ensembles are present in the histograms for all the model components.

**Figure 7 viruses-15-00296-f007:**
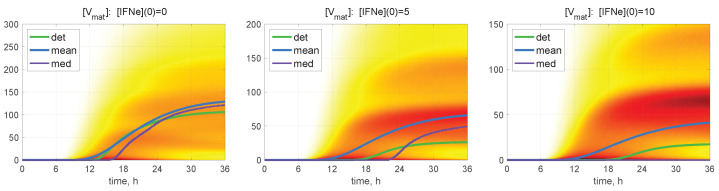
Probability density function (PDF) of stochastic realisations. Time-varying histograms for [V${}_{\mathrm{mat}}$] are computed for MOI=6 and for different values of $\left[{\mathrm{IFN}}_{e}\right]\left(0\right)$ (indicated in the top of every plot). Furthermore, the deterministic solutions (det), mean values (mean), and medians (med) are plotted (colours for the lines are explained in the legend). A darker colour corresponds to a higher value of the histograms. Multi-hump patterns in the trajectory ensembles are present in the histograms.

**Figure 8 viruses-15-00296-f008:**
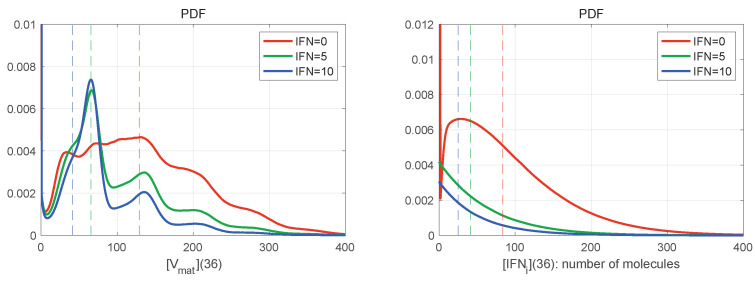
Histograms of mature virions (**left**) and intercellular IFN (**right**) at time t=36 h for MOI=6 and different values of $\left[{\mathrm{IFN}}_{e}\right]\left(0\right)$. The line colours are explained in the legend. The dashed vertical lines indicate the mean values for the corresponding initial extracellular IFN. The histograms are normalised by the number of realisations to approximate the probability distribution function (PDF). They are slightly smoothed by using the Gaussian filter. Both distributions shown are far from the normal distribution.

**Figure 9 viruses-15-00296-f009:**
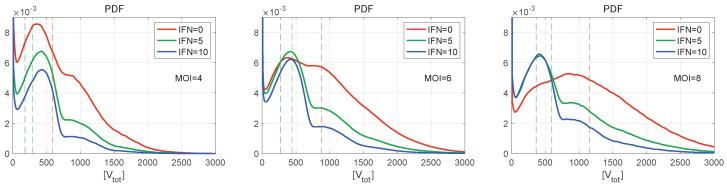
Normalised histograms (PDF) for $\left[V_{\mathrm{tot}}\right]\left(t\right)$ for MOI=4 (**left**), 6 (**centre**), 8 (**right**), and for different values of initial interferon $\left[{\mathrm{IFN}}_{e}\right]\left(0\right)$ (the line colours are explained in the legend). The vertical dashed lines indicate the mean values. The curves demonstrate multiple peaks with the peak amplitude decaying faster with the number of produced virions. The right PDF tails look close to Gaussian distributions. The greater the MOI, the lower is the amplitude of the first peak.

**Figure 10 viruses-15-00296-f010:**
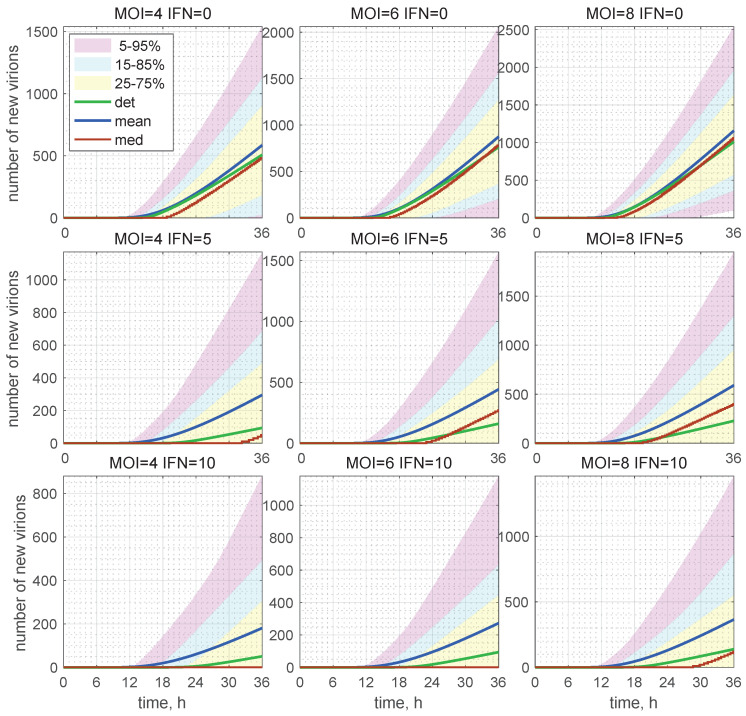
Evolution of the histograms for the total number of released virions during the development of an HIV-1 infection process. Time-varying confidence intervals for [V${}_{\mathrm{new}}$] computed for different MOI and $\left[{\mathrm{IFN}}_{e}\right]\left(0\right)$ indicated in the top of every plot, respectively, as V0 and IFN. Furthermore, the deterministic solutions (det), mean values (mean), and medians (med) are plotted (colours for the patches and lines are explained in the legend). In the presence of extracellular IFN-I, i.e., 10 molecules per cell, the median is identically zero (**left bottom**). This means that, in more than 50% of cases, the stochastic replication process is extinct and new virions are not produced.

**Figure 11 viruses-15-00296-f011:**
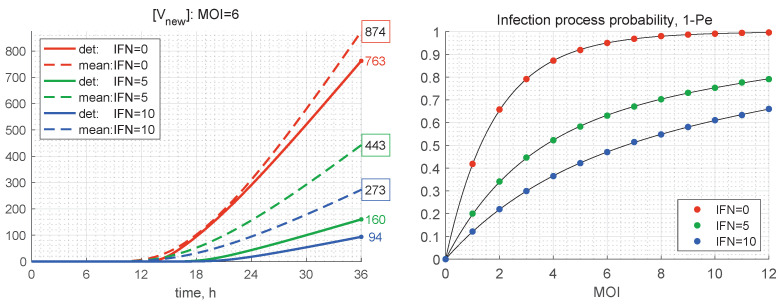
(**Left**) Time variation of the number of produced new virions according to the deterministic model (solid) and its mean value in the stochastic model (dashed) for MOI=6 and different values of $\left[{\mathrm{IFN}}_{e}\right]\left(0\right)$ (the line colours are explained in the legend). The coloured numbers indicate the total number of new virions [V${}_{\mathrm{tot}}$] calculated by the deterministic model for different values of [IFN${}_{e}$]. The black numbers in the coloured boxes indicate the mean values of [V${}_{\mathrm{tot}}$] computed by the stochastic model. (**Right**) Probability for productive infection of a target cell in relation to the initial number of free virions per cell MOI and the initial concentration of extracellular IFN (explained in the legend).

**Table 1 viruses-15-00296-t001:** The Markov chain. Symbols $x_{n}^{--}$ and $x_{n}^{++}$ denote, respectively, the decrease and increase in abundance of component $x_{n}$ by one particle during the *m*th transition.

*m*	Transition	Propensity, $a_{m}$	*m*	Transition	Propensity, $a_{m}$
1	$\;x_{1}^{--},x_{2}^{++}$	$k_{\mathrm{bound}}x_{1}$	42	$x_{18}^{--}$	$d_{\mathrm{Vpu}}x_{18}$
2	$\;x_{1}^{--}$	$d_{\mathrm{free}}x_{1}$	43	$x_{19}^{++}$	$k_{\mathrm{trans}}f_{\mathrm{ss},\mathrm{Vif}}x_{11}$
3	$\;x_{2}^{--},x_{3}^{++}$	$k_{\mathrm{fuse}}x_{2}$	44	$x_{19}^{--}$	$d_{\mathrm{Vif}}x_{19}$
4	$\;x_{2}^{--}$	$d_{\mathrm{bound}}x_{2}$	45	$x_{20}^{--}$	$k_{\mathrm{comb}}N_{\mathrm{Gag}-\mathrm{Pol}}\;F_{c}\;x_{23}$
5	$\;x_{3}^{--},x_{4}^{++}$	$k_{\mathrm{RT}}x_{3}$	46	$x_{20}^{--}$	$d_{\mathrm{mem},\mathrm{Gag}-\mathrm{Pol}}x_{20}$
6	$\;x_{3}^{--}$	$d_{{\mathrm{RNA}}_{\mathrm{cor}}}x_{3}$	47	$x_{21}^{--}$	$k_{\mathrm{comb}}N_{\mathrm{Gag}}\;F_{c}\;x_{23}$
7	$\;x_{3}^{--}$	$f_{\mathrm{APO}}x_{34}x_{3}$	48	$x_{21}^{--}$	$d_{\mathrm{mem},\mathrm{Gag}}x_{21}$
8	$\;x_{3}^{--}$	$f_{\mathrm{SAM}}x_{35}x_{3}$	49	$x_{22}^{--}$	$k_{\mathrm{comb}}N_{\mathrm{gp}160}\;F_{c}\;x_{23}$
9	$\;x_{4}^{--},x_{5}^{++}$	$k_{{\mathrm{DNA}}_{t}}x_{4}$	50	$x_{22}^{--}$	$d_{\mathrm{mem},\mathrm{gp}160}x_{22}$
10	$\;x_{4}^{--}$	$d_{{\mathrm{DNA}}_{\mathrm{cor}}}x_{4}$	51	$x_{23}^{--}$	$k_{\mathrm{comb}}N_{\mathrm{RNA}}\;F_{c}\;x_{23}$
11	$\;x_{5}^{--},x_{6}^{++}$	$k_{\mathrm{int}}x_{5}$	52	$x_{23}^{--}$	$d_{\mathrm{RNA}}x_{23}$
12	$\;x_{5}^{--}$	$d_{{\mathrm{DNA}}_{\mathrm{nuc}}}x_{5}$	53	$x_{24}^{--}$	$d_{\mathrm{Vpu}}x_{24}$
13	$\;x_{6}^{--}$	$d_{{\mathrm{DNA}}_{\mathrm{cor}}}x_{4}$	54	$x_{25}^{++}$	$k_{\mathrm{comb}}\;F_{c}\;x_{23}$
14	$\;x_{7}^{++}$	$f_{\mathrm{TR}}x_{6}$	55	$x_{25}^{--},x_{26}^{++}$	$k_{\mathrm{bud}}x_{25}$
15	$\;x_{7}^{--},x_{8}^{++}$	$k_{\mathrm{eRNA}}f_{\mathrm{Rev}}x_{7}$	56	$x_{25}^{--}$	$d_{\mathrm{comb}}x_{25}$
16	$\;x_{7}^{--},x_{10}^{++}$	$k_{{\mathrm{ssRNA}}_{g}}g_{\mathrm{Rev}}x_{7}$	57	$x_{26}^{--},x_{27}^{++}$	$k_{\mathrm{bud}}x_{26}$
17	$\;x_{7}^{--}$	$d_{{\mathrm{DNA}}_{\mathrm{cor}}}x_{7}$	58	$x_{26}^{--}$	$d_{\mathrm{bud}}x_{26}$
18	$\;x_{8}^{--},x_{9}^{++}$	$k_{{\mathrm{dsRNA}}_{\mathrm{ss}}}g_{\mathrm{Rev}}x_{8}$	59	$x_{26}^{--}$	$f_{\mathrm{Teth}}x_{36}x_{26}$
19	$\;x_{8}^{--},x_{11}^{++}$	$k_{{\mathrm{eRNA}}_{\mathrm{ss}}}g_{\mathrm{Rev}}x_{8}$	60	$x_{27}^{--}$	$d_{\mathrm{free}}x_{27}$
20	$\;x_{8}^{--}$	$d_{{\mathrm{RNA}}_{\mathrm{ss}}}x_{8}$	61	$x_{28}^{++}$	$k_{\mathrm{RIG}1}x_{3}$
21	$\;x_{9}^{--},x_{12}^{++}$	$k_{{\mathrm{eRNA}}_{\mathrm{ds}}}g_{\mathrm{Rev}}x_{8}$	62	$x_{28}^{--}$	$d_{\mathrm{RIG}}x_{28}$
22	$\;x_{9}^{--}$	$d_{{\mathrm{RNA}}_{\mathrm{ds}}}x_{9}$	63	$x_{29}^{++}$	$k_{\mathrm{IRF}3}x_{28}$
23	$x_{10}^{--},x_{23}^{++}$	$k_{\mathrm{tp},\mathrm{RNA}}x_{10}$	64	$x_{29}^{--}$	$d_{\mathrm{IRF}3}x_{29}$
24	$x_{10}^{--}$	$d_{{\mathrm{RNA}}_{g}}x_{10}$	65	$x_{30}^{++}$	$k_{\mathrm{NF}-\mathrm{kb}}x_{28}$
25	$x_{11}^{--}$	$d_{{\mathrm{RNA}}_{\mathrm{ss}}}x_{11}$	66	$x_{30}^{--}$	$d_{\mathrm{NF}-\mathrm{kb}}x_{30}$
26	$x_{12}^{--}$	$d_{{\mathrm{RNA}}_{\mathrm{ds}}}x_{12}$	67	$x_{31}^{++}$	$k_{\mathrm{INFi}}x_{29}$
27	$x_{13}^{++}$	$k_{\mathrm{trans}}f_{\mathrm{ds},\mathrm{Tat}}x_{12}$	68	$x_{31}^{++}$	$k_{\mathrm{INFi}}x_{30}$
28	$x_{13}^{--}$	$d_{\mathrm{TaT}}x_{13}$	69	$x_{31}^{--},x_{32}^{++}$	$k_{e}x_{31}$
29	$x_{14}^{++}$	$k_{\mathrm{trans}}f_{\mathrm{ds},\mathrm{Rev}}x_{12}$	70	$x_{31}^{--}$	$d_{\mathrm{INFi}}x_{31}$
30	$x_{14}^{--}$	$d_{\mathrm{TaT}}x_{14}$	71	$x_{32}^{--}$	$d_{\mathrm{INFe}}x_{32}$
31	$x_{15}^{++}$	$k_{\mathrm{trans}}f_{g,\mathrm{Gag}-\mathrm{Pol}}x_{10}$	72	$x_{33}^{++}$	$k_{\mathrm{STAT}}x_{32}$
32	$x_{15}^{--},x_{20}^{++}$	$k_{\mathrm{tp},\mathrm{Gag}-\mathrm{Pol}}x_{15}$	73	$x_{33}^{--}$	$d_{\mathrm{STAT}}x_{33}$
33	$x_{15}^{--}$	$d_{\mathrm{Gag}-\mathrm{Pol}}x_{15}$	74	$x_{34}^{++}$	$k_{\mathrm{ISG}}x_{33}$
34	$x_{16}^{++}$	$k_{\mathrm{trans}}f_{g,\mathrm{Gag}}x_{10}$	75	$x_{34}^{--}$	$f_{\mathrm{Vif}}x_{20}x_{34}$
35	$x_{16}^{--},x_{21}^{++}$	$k_{\mathrm{tp},\mathrm{Gag}}x_{16}$	76	$x_{34}^{--}$	$d_{\mathrm{APO}}x_{34}$
36	$x_{16}^{--}$	$d_{\mathrm{Gag}}x_{16}$	77	$x_{35}^{++}$	$k_{\mathrm{ISG}}x_{33}$
37	$x_{17}^{++}$	$k_{\mathrm{trans}}f_{\mathrm{gp}160}x_{11}$	78	$x_{35}^{--}$	$d_{\mathrm{SAMHD}1}x_{35}$
38	$x_{17}^{--},x_{22}^{++}$	$k_{\mathrm{tp},\mathrm{gp}160}x_{17}$	79	$x_{36}^{++}$	$k_{\mathrm{ISG}}x_{33}$
39	$x_{17}^{--}$	$d_{\mathrm{gp}160}x_{10}$	80	$x_{36}^{--}$	$f_{\mathrm{Vpu}}x_{19}x_{36}$
40	$x_{18}^{++}$	$k_{\mathrm{trans}}f_{\mathrm{ss},\mathrm{Vpu}}x_{11}$	81	$x_{36}^{--}$	$d_{\mathrm{Teth}}x_{36}$
41	$x_{18}^{--},x_{24}^{++}$	$k_{\mathrm{tp}}x_{18}$			

Here, *f_TR_* and *F_c_* are calculated by Equations (12a) and (25a), respectively.

**Table 2 viruses-15-00296-t002:** The number of total virions released from a single infected cell during the infection process for different values of [V${}_{\mathrm{free}}$](0) (MOI) and [IFN${}_{e}$](0) (molecules). Here, “det” denotes the outputs in the deterministic process, “mean” denotes the mean values in the stochastic process, and Δ means the excess of the mean stochastic outputs over the deterministic outputs in percent. The “Life Cycle Efficiency” is defined by Equation (40). “IFN Inhibitory Factor” is defined by Equation (41).

	MOI=4			MOI=6			MOI=8		
	det	mean	Δ	det	mean	Δ	det	mean	Δ
$\left[{\mathrm{IFN}}_{e}\right]\left(0\right)=0$	508	585	15%	763	874	15%	1010	1158	15%
Life Cycle Efficiency	127	146		127	146		126	145	
$\left[{\mathrm{IFN}}_{e}\right]\left(0\right)=5$	94	295	215%	160	443	176%	228	591	159%
Life Cycle Efficiency	23	74		27	74		28	74	
IFN Inhibitory Factor	5.42	1.98		4.76	1.98		4.43	1.96	
$\left[{\mathrm{IFN}}_{e}\right]\left(0\right)=10$	51	181	257%	94	273	191%	365	163	49%
Life Cycle Efficiency	13	45		16	45		17	46	
IFN Inhibitory Factor	10.0	3.24		8.31	3.21		7.29	3.17	

## Data Availability

Not applicable.

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
