# Peer review of "Stochastic Modelling of HIV-1 Replication in a CD4 T Cell with an IFN Response"

_viruses, 2023, doi:10.3390/v15020296_

Round 1
Reviewer 1 Report
A major revision is required for the research article.

Author Response
Response to comments of Reviewer 1
In this research article, the authors have studied mathematical model describing the HIV-1 life 18 cycle in CD4 T cells. The deterministic and the stochastic models are formulated and the investigations explored the activation of the intracellular type I interferon (IFN-I) response and the IFN-induced suppression of viral replication. There are a few points that I want to mention and I think the authors should address these very carefully.
Overall, the main idea of the paper is interesting. The whole manuscript should be revised by the authors and the necessary changes and corrections should essentially be done by the authors according to the points mentioned above. Major revision is strictly required for further consideration of the manuscript in the journal ‘Viruses, MDPI’.
We thank the Reviewer for insightful comments and the thorough work on our manuscript.
There are a few points that I want to mention and I think the authors should address these very carefully.
- The introduction section needs to be more informative. More literature review is needed.
Our reply: In response to the suggestion, we have added a paragraph reviewing the importance of interferon system in the pathogenesis of HIV-1 infection.
- What is the motivation of the work? Discuss clearly the novelty and innovativeness of this research work.
Our reply: In response to the suggestion, we have added a paragraph discussion the motivation of this research work.
- Figure captions needs to be more elaborative.
Our reply: The legends to figures have been additionally elaborated to provide more details.
- The ‘Discussion’ and ‘Conclusion’ sections are not well-written. More clinical and experimental results from the recent studies should be compared with obtained interpretations.
Our reply: In response to the concern of the reviewer, we have added paragraphs to Discussion and Conclusion section highlighting the clinical relevance and novelty of the results of the study.
- Correct all the typos and grammatical mistakes.
Our reply: All found typos and grammatical mistakes have been corrected.
- I suggest being precise whenever describing the methodology portion. The less important parts should be moved to the Appendix section.
Our reply: The methods section presents the derivation of the model equations, specification of the model parameters and the approach to building the stochastic model. The provided details of the model are necessary for the readers to fully understanding the underlying assumptions as well as the limitations of the mechanistic approach. All equations are equally important for a comprehensive perception of the model and none of them is worth to move to Appendix.
- Numerical simulations are not so satisfying. Much more variety of enriched Figures should be incorporated to justify or validate the achieved outcomes.
Our reply: The above remark is not specific enough. Our study already includes 11 figures and 2 Tables. The achieved outcomes of the study are illustrated by Figures 2-11 and discussed in subsections 3.1 to 3.7 of Section 3. We would be happy to add more figures once the suggestion of the reviewer is formulated in a more clear way.

Reviewer 2 Report
1. Line 86-89 "Note that as a result of the interaction of interferon...yield of virions by 40% [11]." Is this referring to prior experimental observations or results from the model? Should clarify.
2. Line 271 "3.1. Sensitivity of HIV-1 and IFN-I secretion to IFN-mediated control" I think the I is incorrect on IFN-I
3. For section 3.1, it should be described why wasn't the sensitivity of the model to other parameters discussed. Especially as this was done with the deterministic form, that could be done very quickly.
4. For section 3.2, are the simulations initiated with the same initial conditions or are they randomized as well? If they are starting with the same initial conditions, please state that in this paragraph. Makes it more clear that the differences observed in Fig 4. are due to stochasticity only.
5. Move Fig 5 above Fig 4. Improves readability and then can more easily understand the color map.
After reading the work, while I really enjoy a nice study comparing stochastic and deterministic models, I do not see much gained here in terms of our understanding of the infection; or more specifically, the interplay between IFN and HIV replication. The conclusions are not focused on the infection. I would recommend this be published in another MDPI journal with a more appropriate audience. Specifically Processes.
Author Response
Response to comments of Reviewer 2
We thank the Reviewer for insightful comments and the thorough work on our manuscript.
- Line 86-89 "Note that as a result of the interaction of interferon...yield of virions by 40% [11]." Is this referring to prior experimental observations or results from the model? Should clarify.
Our reply: Paper [11] ([21] in the revised version) is experimental. The conclusion is based on observation. It is indicated now in the paper.
- Line 271 "3.1. Sensitivity of HIV-1 and IFN-I secretion to IFN-mediated control" I think the I is incorrect on IFN-I.
Our reply: Both notations: “IFN-I” and “IFN-1”, are used for type-I interferon. We prefer to use notation “IFN-I” as in majority of our referenced papers.
- For section 3.1, it should be described why wasn't the sensitivity of the model to other parameters discussed. Especially as this was done with the deterministic form, that could be done very quickly.
Our reply: The focus of our study is on the analysis of type I IFN response effect on HIV-1 replication. The study is built upon our previous work [Shcherbatova O., et al., Modeling of the HIV-1 Life Cycle in Productively Infected Cells to Predict Novel Therapeutic Targets. Pathogens. 2020; 9(4):255], in which the sensitivity of HIV-1 life cycle to all considered biochemical reaction steps was examined in full. In order, to keep the focus of the present work on IFN-I function, we restrict the sensitivity examination using heatmaps, uncertainty bands, and pdfs to characterize the IFN-I secretion and IFN-I-induced function of ISG-produced proteins inhibiting the virus replication. The respective comment has been added to Subsection 3.1.
- For section 3.2, are the simulations initiated with the same initial conditions or are they randomized as well? If they are starting with the same initial conditions, please state that in this paragraph. Makes it more clear that the differences observed in Fig 4. are due to stochasticity only.
Our reply: At the end of Section 2.1, we describe clearly the initial conditions:
“At time t = 0, the initial conditions of infection scenario are set as follows: initial number of free virions, x1(0) = [Vfree](0), i.e MOI and the initial number of extracellular IFN: x31(0) = [IFNe](0) molecules. All other components are zeros at t = 0.”
Thus, the only nonzero initial conditions are MOI and the number of extracellular IFN molecules. They are taken fixed for every series of runs. All non-zero initial values used for numerical simulations are presented in Table 2: MOI=4,6,8, [IFNe]=0,5,10.
- Move Fig 5 above Fig 4. Improves readability and then can more easily understand the color map.
Our reply: If we swap Figures 4 and 5 it would destroy logic of the paper. We present first the plots for the main variables participating in the process and described by ODE (1)-(36), and only after that we put the plots for auxiliary values computed on the base of the main variables. We think that the curve colors are explained in Figure 5 following Figure 4.
After reading the work, while I really enjoy a nice study comparing stochastic and deterministic models, I do not see much gained here in terms of our understanding of the infection; or more specifically, the interplay between IFN and HIV replication. The conclusions are not focused on the infection. I would recommend this be published in another MDPI journal with a more appropriate audience. Specifically Processes.
Our reply: We suppose that the revised paper, in which all comments have been addressed, fits well into Viruses’s area of interest.

Reviewer 3 Report
Authors propose stochastic modeling of HIV-1 replication in a CD4 T cell with an IFN response. I find the draft well written and with good ideas. However, I have some questions:
* There is a tone of parameters, and they are just taken from previous work. It would be nice to have in the paper how these parameters were obtained.
* What is the difference between this paper and your previous draft in Pathogens?
* I suggest to label equations only with numbers.
* It is difficult to find the impact of the model to the biological problem.
* Is it needed to have all these equations for such a simple dybamics?
Author Response
Response to comments of Reviewer 3
We thank the Reviewer for insightful comments and the thorough work on our manuscript.
- There is a tone of parameters, and they are just taken from previous work. It would be nice to have in the paper how these parameters were obtained.
Our reply: The parameters of the model can be subdivided into two subsets, the first one refers to the biochemical reactions of HIV-1 components replication and the second set describes the IFN-I related reactions. Please, note that the sources of IFN-I-related parameters are provided at the end of Subsection 2.2. Whereas specific details for HIV-1 life cycle parameters are presented in our previous study [Shcherbatova O., et al., Modeling of the HIV-1 Life Cycle in Productively Infected Cells to Predict Novel Therapeutic Targets. Pathogens. 2020; 9(4):255], in response to the suggestion of the Reviewer, we added the following brief overview of their calibration after the description of the model equations in Subsection 2.2:
“In particular, the reaction rate constants of HIV-1 transcription, translation and assembly stages have been estimated using two sources of data, i.e., the previously formulated mathematical models for the respective processes and the experimental data coming from quantitative studies of HIV-1 life cycle stages. One of the key experimental studies which provides a temporal description of HIV-1 replication [Mohammadi, P. et al., A. 24 Hours in the Life of HIV-1 in a T Cell Line. PLoS Pathog. 2013, 9, e1003161.] has been used to refine the estimates of some parameters. It also describes the kinetics of the RNA-, DNA-, and viral proteins and therefore served for an overall validation of our model-reproduced dynamics.”
- I suggest to label equations only with numbers?
Corrected
- It is difficult to find the impact of the model to the biological problem.
Our reply: We suppose that the effect of initial extracellular IFN on the infection process in cell is an essential biological problem important for find adequate treatment methods. We have added some sentences and paragraphs in Introduction and Conclusion to clarify this.
- Is it needed to have all these equations for such a simple dynamics?.
Our reply: The number of equations is determined by the number of all species participating in the infection process. The equations are based on chemical kinetics methods but, along with molecules, other particles, like virions and their fragments, participate. To understand the infection process and factors affecting it, we have to know its dynamics in detail. We would not call the infection process in a cell “a simple dynamics”.

Round 2
Reviewer 3 Report
I think the work is OK for publication.